# Role of Phosphodiesterase 1 in the Regulation of Real-Time cGMP Levels and Contractility in Adult Mouse Cardiomyocytes

**DOI:** 10.3390/cells12232759

**Published:** 2023-12-03

**Authors:** Nadja I. Bork, Hariharan Subramanian, Roberta Kurelic, Viacheslav O. Nikolaev, Sergei D. Rybalkin

**Affiliations:** 1Institute of Experimental Cardiovascular Research, University Medical Center Hamburg-Eppendorf, 20246 Hamburg, Germany; n.bork@uke.de (N.I.B.); h.subramanian@uke.de (H.S.); r.kurelic@uke.de (R.K.);; 2German Center for Cardiovascular Research (DZHK), Partner Site Hamburg/Kiel/Lübeck, 20246 Hamburg, Germany

**Keywords:** PDE1, cGMP, cardiomyocyte, FRET

## Abstract

In mouse cardiomyocytes, the expression of two subfamilies of the calcium/calmodulin-regulated cyclic nucleotide phosphodiesterase 1 (PDE1)—PDE1A and PDE1C—has been reported. PDE1C was found to be the major subfamily in the human heart. It is a dual substrate PDE and can hydrolyze both 3′,5′-cyclic adenosine monophosphate (cAMP) and 3′,5′-cyclic guanosine monophosphate (cGMP). Previously, it has been reported that the PDE1 inhibitor ITI-214 shows positive inotropic effects in heart failure patients which were largely attributed to the cAMP-dependent protein kinase (PKA) signaling. However, the role of PDE1 in the regulation of cardiac cGMP has not been directly addressed. Here, we studied the effect of PDE1 inhibition on cGMP levels in adult mouse ventricular cardiomyocytes using a highly sensitive fluorescent biosensor based on Förster resonance energy transfer (FRET). Live-cell imaging in paced and resting cardiomyocytes showed an increase in cGMP after PDE1 inhibition with ITI-214. Furthermore, PDE1 inhibition and PDE1A knockdown amplified the cGMP-FRET responses to the nitric oxide (NO)-donor sodium nitroprusside (SNP) but not to the C-type natriuretic peptide (CNP), indicating a specific role of PDE1 in the regulation of the NO-sensitive guanylyl cyclase (NO-GC)-regulated cGMP microdomain. ITI-214, in combination with CNP or SNP, showed a positive lusitropic effect, improving the relaxation of isolated myocytes. Immunoblot analysis revealed increased phospholamban (PLN) phosphorylation at Ser-16 in cells treated with a combination of SNP and PDE1 inhibitor but not with SNP alone. Our findings reveal a previously unreported role of PDE1 in the regulation of the NO-GC/cGMP microdomain and mouse ventricular myocyte contractility. Since PDE1 serves as a cGMP degrading PDE in cardiomyocytes and has the highest hydrolytic activities, it can be expected that PDE1 inhibition might be beneficial in combination with cGMP-elevating drugs for the treatment of cardiac diseases.

## 1. Introduction

Cyclic nucleotide phosphodiesterases (PDEs) play an essential role in cyclic nucleotide signaling by degrading 3′,5′-cyclic adenosine monophosphate (cAMP) and 3′,5′-cyclic guanosine monophosphate (cGMP). A total of 21 genes encode for 11 PDE superfamilies (PDE1-PDE11) with >100 isoforms [1,2,3].

The calcium (Ca^2+^)/calmodulin regulated PDE1 is a dual substrate PDE, degrading cAMP and cGMP. Of the three existing PDE1 subfamilies, PDE1A and PDE1C are expressed in heart and vessels, PDE1B is primarily expressed in the brain. The three subfamilies vary in their cyclic nucleotide affinity. Whereas PDE1A is >20 times more selective for cGMP, and PDE1B also preferentially hydrolyzes cGMP, PDE1C hydrolyzes both nucleotides equally well [3,4]. In larger mammalian and human hearts, PDE1C is predominately expressed, whereas PDE1A is the dominant isoform expressed in small rodents [4,5,6].

Recently, the potent and highly selective PDE1 inhibitor ITI-214 has been developed [7]. It was first explored in humans for neurocognitive diseases (NCT01900522 and NCT03257046). More recently, it was studied in patients with heart failure and reduced ejection fraction, where it showed positive inotropic effects (NCT03387215) [8]. These inotropic–vasodilator effects in patients were similar to those found in a study performed on dogs and rabbits by Hashimoto et al. [4], where they were connected to cAMP signaling. Recently, Muller at al. [5] demonstrated in primary guinea pig myocytes that the positive inotropic effects of PDE1 inhibition with ITI-214 were attributable to cAMP-dependent protein kinase A (PKA) signaling. In all those studies, the contraction responses to ITI-214 were shown to be PKA-dependent, suggesting that cAMP is the major downstream signaling regulator. However, the precise localization of PDE1-regulated cAMP microdomains has not been shown [5]. Furthermore, the regulation of cardiac cGMP has not been directly addressed and the potential role for cGMP remains unexplored.

Here, we tested the hypothesis that cardiac PDE1 can also regulate specific cGMP microdomains. We have so far been able to demonstrate an unknown role of PDE1 in the regulation of the nitric oxide (NO)-sensitive guanylyl cyclase (NO-GC) cGMP microdomains and mouse ventricular myocyte contractility.

## 2. Materials and Methods

### 2.1. Chemicals and Kits

CNP was purchesed from Bachem (Bubendorf, Switzerland), Fura-2 (F1221) from Thermo Scientific (Waltham, MA, USA), IBMX from AppliChem (Darmstadt, Germany), ITI 214 from MedChemExpress (Monmouth Junction, NJ, USA), phosphatase and protease inhibitors from Roche (Basel, Switherland), and SNP and all other chemicals from Sigma-Aldrich (Deisenhofen, Germany).

### 2.2. Animals

Animal work with both female and male mice aged between 8 and 20 weeks was carried out according to national and international animal welfare guidelines and approved by the local animal welfare authority of BGJ Hamburg (approval number ORG_1010, ORG_1113).

For cGMP-FRET measurements, transgenic mice expressing the cGMP biosensor red-cGES-DE5 under the control of the α-myosin heavy chain (αMHC) promoter on the C57BL6/J background [9] were used.

For PDE3 knockout (KO), mice with the cardiomyocyte-specific conditional knockout of PDE3A (PDE3A^d/d^Cre^+^) on the C57BL6/J background [9] were used.

### 2.3. Cardiomyocyte Isolation

Adult ventricular cardiomyocytes were isolated as described previously [9].

### 2.4. Live Cell Imaging

FRET measurements were carried out as previously described [9]. Briefly, isolated cardiomyocytes plated onto laminin-coated coverslips were excited with 400 nm LED (pE-100). A DV2 Dual View beamsplitter with a 565 dcxr dichroic mirror and 520/30 and D630/50 emission filters (Teledyne Photometrics, Tucson, AZ, USA) was used to split emission light into donor and acceptor channels. An OptiMOS CMOS camera (Teledyne Photometrics, Tucson, AZ, USA) was used for recording. Measurements were performed at room temperature, and images were taken every 5 s.

For pacing experiments, cells were stimulated at 1 Hz and 20 V with a Myopacer cell stimulator (IonOptix, Westwood, MA, USA) as previously described [10]. After 3 min of the equilibration period, FRET measurements were begun.

Micro Manager 1.4.5 software was used for image acquisition.

### 2.5. IonOptix

Freshly isolated cardiomyocytes were plated on laminin-coated glass-bottom chambers and incubated at 37 °C for an hour before Fura-2 loading (5 µM, 20 min). After several washes, cells were field-stimulated at 1 Hz and 20 V. In the paced cells, contraction/relaxation parameters and calcium transients, including relaxation and Ca^2+^ reuptake values (T_50_), were analyzed in >10 cells/mice using high-throughput IonOptix system.

### 2.6. Cardiomyocyte Stimulation and Immunoblot Analysis

Freshly isolated cardiomyocytes were stimulated for 10 min at room temperature in 1.5 mL reaction tubes. After harvesting, cells were lysed in RIPA buffer (150 mM NaCl, 1% triton, 0.1% SDS, 0.5% sodium deoxycholate, 50 mM tris pH 8.0, and phosphatase and protease inhibitors and then sonicated (3 × 15 s, 30%). After adding 4× Laemmli loading dye, samples were boiled for 5 min at 95 °C. Proteins were size separated on 15% polyacrylamide gels using SDS polyacrylamide gel electrophoresis (SDS-PAGE) and transferred onto PVDF membrane (Biorad, Feldkirchen, Germany) using the tank blot method.

For immunodetection, P-PLN (dilution 1:5000, Badrilla A010-12) and T-PLN (dilution 1:2500, abcam 126174) were used as specific primary antibodies. Images were analyzed with ImageJ 1.44 software.

### 2.7. siRNA-Mediated Silencing of PDE1

For siRNAs, on-target plus siRNA SMART pools from Dharmacon (mouse PDE1A, L-047396-01-0005, and mouse PDE1C, L-041116-01-0005) were used.

For transfection with siRNA, freshly isolated cardiomyocytes were plated on laminin-coated coverslips. At 2 h after plating, cardiomyocytes were transfected with siRNA (25 nM siRNA per well) using DharmaFECT transfection reagent (Dharmacon, Horizon Discovery, Waterbeach, UK, #T-2001-02), according to the manufacturer’s instructions. Cells were used for FRET experiments 48 h after transfection.

### 2.8. Statistics

GraphPad Prism 9 (version 9.3.0) was used for statistical analysis. Data are presented as mean ± SEM. Normal distribution was tested via D’Agostino’s and Pearson’s tests. For a small sample number (*n* < 8), normal distribution was tested with the Kolmogorov–Smirnov test. Normally distributed data were analyzed using an unpaired or paired *t*-test, or one-way ANOVA followed by Sidak’s multiple comparisons test. Measurements taken from several cells from different mice were assessed with a nested *t*-test or nested ANOVA. Values of *p* < 0.05 were considered statistically significant.

## 3. Results

To analyze the effect of PDE1 inhibition on cGMP levels in adult mouse ventricular cardiomyocytes, we performed cGMP live cell imaging in cells expressing the FRET-based cGMP biosensor red-cGES-DE5 [9]. Live-cell FRET measurements showed an increase in intracellular cGMP in response to the selective PDE1 inhibitor ITI-214 applied alone (Figure 1A,C).

PDE1 is a Ca^2+^/calmodulin regulated PDE and the binding of Ca^2+^/calmodulin to PDE1 stimulates cyclic nucleotide hydrolysis [3]. To assess how Ca^2+^ fluctuations during excitation–contraction coupling in cardiomyocytes impact PDE1 activity, we measured cGMP-FRET responses to PDE1 inhibition in paced cardiomyocytes. Therefore, we used an electric field stimulation protocol which leads to consistent Ca^2+^ transients but does not interfere with normal FRET measurements [10]. Treatment with ITI-214 alone showed relatively small cGMP-FRET responses with no significant differences in paced cardiomyocytes as compared to resting cells (Figure 1B,C).

Cardiac cGMP is compartmentalized in several subcellular microdomains by local pools of PDEs and GCs, including NO-GC and membrane-bound or particulate GCs, which are receptors for natriuretic peptides [11,12]. To analyze the effect of PDE1 on the particulate GC-generated cGMP, we performed cGMP-FRET measurements using PDE1 inhibitor ITI-214 after prestimulation with submaximal concentrations of the C-type natriuretic peptide CNP. In resting myocytes, CNP stimulation raised cytosolic cGMP as expected. The subsequent addition of ITI-214 further increased cGMP levels. Additionally, we treated the cells with the pan-PDE inhibitor IBMX to induce maximal cGMP-FRET response (Figure 2A). To investigate whether electric field stimulation has any effect on cGMP-FRET response to ITI-214 after prestimulation with CNP, we performed the same FRET protocol in paced cardiomyocytes. Our measurements showed no significant differences in cGMP-FRET response to ITI-214 after CNP prestimulation between resting and paced cardiomyocytes (Figure 2A,C).

Next, we investigated whether PDE1 plays a direct role in the regulation of cGMP generated by particulate GC. Therefore, we measured the CNP-induced cGMP-FRET response in resting and paced cardiomyocytes after the inhibition of PDE1 with ITI-214, followed by IBMX (Figure 2B). The quantification of cytosolic FRET responses revealed no significant change in CNP-induced cGMP-FRET response following PDE1 inhibition either in resting or in paced cardiomyocytes compared to CNP response without PDE1 preinhibition (Figure 2B,C). Interestingly, the ITI-214 response after CNP was significantly increased compared to ITI-214 without CNP prestimulation in both resting and paced cells (Figure 2C).

Cardiomyocyte cGMP can also be generated by NO-sensitive GC (NO-GC). Therefore, we further investigated the effect of PDE1 on NO-GC/cGMP signaling using the NO-donor sodium nitroprusside (SNP) to stimulate NO-GC and measured cGMP-FRET response to ITI-214 after SNP-prestimulation in resting and paced cardiomyocytes. As already shown in several studies [13,14], SNP alone only led to a negligible increase in cGMP-FRET in both resting and paced cardiomyocytes (Figure 3A,C). ITI-214 after SNP raised cytosolic cGMP in resting and paced myocytes, which can be further increased by IBMX. In paced cells, the ITI-214 response after SNP was significantly increased compared to resting cells (Figure 3A,C).

Next, we investigated whether PDE1 plays a direct role in the regulation of the NO-GC/cGMP pool. Therefore, we performed cGMP-FRET measurements where we first blocked PDE1 with ITI-214, which raised cGMP levels, as already shown previously (Figure 1 and Figure 2), and then stimulated the cells with the NO-donor SNP followed by IBMX. Strikingly, SNP after ITI-214 led to a ~2-fold increase in cGMP-FRET compared to SNP alone in both resting and paced myocytes (Figure 3B,C). These findings suggest a functional role of PDE1 in the regulation of the NO-GC/cGMP microdomain. In paced myocytes, ITI-214 response after SNP was significantly increased compared to ITI-214 without SNP prestimulation (Figure 3C).

cGMP/cGMP-dependent protein kinase I (PKG also known as cGKI) mediated the phosphorylation of phospholamban (PLN), and subsequent activation of the sarcoplasmic reticulum Ca^2+^-ATPase (SERCA2a) activity exerted positive lusitropic effects in rodent hearts [15,16,17]. Hence, we further investigated the effect of PDE1 inhibition in combination with CNP or SNP on Ca^2+^ transients (Figure 4A,B), Ca^2+^ re-uptake kinetics (Figure 4C) and sarcomere shortening (Figure 4D,E) and relaxation (T_50_) (Figure 4F) in isolated myocytes. Interestingly, ITI-214, in combination with both CNP and SNP, but not alone, showed clear positive lusitropic effects, improving the relaxation of isolated myocytes (Figure 4).

PKG can increase SERCA2a activity through the phosphorylation of PLN at Ser-16, thereby increasing cytosolic Ca^2+^ re-uptake [15,16,18]. In order to test whether PDE1 inhibition affects PLN phosphorylation (at Ser-16) after stimulation of pGC/cGMP/cGKI- or NO-GC/cGMP/cGKI-signaling axis, we performed the immunoblot analysis of myocytes treated with a combination of CNP and PDE1 inhibitor and CNP alone or SNP and PDE1 inhibitor and SNP alone. As expected, cells treated with CNP showed a significant increase in PLN phosphorylation (Ser-16) compared to untreated controls. Also, ITI-214 alone slightly but significantly increased PLN phosphorylation (Ser-16) as compared to the control. The combination of CNP and ITI-214 further increased the phosphorylation of PLN compared to CNP or ITI-214 alone (Figure 5A). Interestingly, whereas SNP alone did not increase PLN phosphorylation (Ser-16), the combination of SNP and ITI-214 significantly increased the phosphorylation of PLN (Ser-16) compared to untreated control and ITI-214 alone (Figure 5B).

ITI-214 is a picomolar PDE inhibitor with excellent selectivity over other PDE families [7]. Its inhibitory constant (PDE1A K_i_ = 34 pmol; PDE1B K_i_ = 380 pmol; PDE1C K_i_ = 37 pmol) has >1000-fold selectivity for the nearest other PDE family (PDE4D K_i_ = 33 nM) and 10,000–300,000-fold selectivity towards all other PDE enzyme families [19].

In murine cardiomyocytes, PDE3A is the main PDE responsible for the regulation of basal and stimulated cGMP degradation [9], and also in human cardiomyocytes, PDE3A is the most abundant PDE expressed [20]. PDE3A was shown to control PLN-SERCA2a activity by acting in a SERCA2a-PLN-A-kinase anchoring protein 18 (AKAP18) multiprotein complex [21,22,23]. In order to ensure that the increase in PLN phosphorylation by ITI-214 (as seen in Figure 5) was not caused by ITI-214 affecting PDE3A activity, we used cardiomyocytes from mice with a cardiomyocyte-specific deletion of PDE3A using the Cre/loxP system [9] and then analyzed PLN phosphorylation. Immunoblot analysis in PDE3A^d/d^Cre^+^ cardiomyocytes revealed increased PLN phosphorylation at Ser-16 in cells treated with either CNP, ITI-214, or a combination of both (Figure 6). Cells treated with a combination of SNP and PDE1 inhibitor but not with SNP alone showed increased PLN phosphorylation at Ser-16 compared to untreated controls and cells treated with ITI-214 alone (Figure 6). This supports our hypothesis that PDE1 is involved in the regulation of particulate GC/cGMP/PKG-signaling.

In order to address the question of which PDE1 subfamily—PDE1A or PDE1C—is responsible for the regulation of the NO-GC/cGMP microdomain, we selectively silenced PDE1A, PDE1C, or both subfamilies together in mouse cardiomyocytes using siRNA and performed cGMP-FRET measurements (Figure 7, Appendix A). Interestingly, the knockdown of PDE1A significantly reduced cGMP-FRET response to ITI-214 after stimulation with NO-donor SNP (Figure 7B,E). Whereas silencing PDE1C alone had no significant effect on ITI-214 FRET response, the combined silencing of PDE1A and PDE1C showed an even greater, almost complete inhibition of ITI-214 response after SNP (Figure 7C,E). This indicates that both PDE1A and PDE1C together regulate the NO-GC/cGMP microdomain in mouse cardiomyocytes.

## 4. Discussion

In the present investigation, we used a FRET-based cGMP live-cell imaging approach with a highly sensitive sensor red-cGES-DE5 to show that PDE1 inhibition by ITI-214 could unmask the effect of the NO-GC/cGMP pathway in adult mouse ventricular myocytes (Figure 1, Figure 2 and Figure 3). To the best of our knowledge, this is the first time that a direct, immediate effect of PDE1 inhibition on cGMP levels stimulated by NO donors in adult murine ventricular myocytes could be shown.

Previously, ITI-214 was used in several studies to investigate its effect on heart failure in heart failure models [4,5,8]. In the first publication, ITI-214 was administered to heart failure dogs and rabbits to analyze pressure–volume relationships in vivo [4]. This treatment resulted in acute inotropic, lusitropic, and arterial vasodilatory effects. Interestingly, unlike inotropic effects from β-adrenergic receptor stimulation or PDE3 inhibition, these inotropic effects were not accompanied by an increase in Ca^2+^ transient amplitude. However, they occurred mainly via cAMP modulation coupled to adenosine A_2B_ receptor (A_2B_R) signaling rather than β-adrenergic signaling. At the same time, it was also shown that ITI-214 was able to elevate cAMP in isolated rabbit cardiomyocytes in the presence of forskolin when compared with forskolin alone. Later, as the possible mechanism to explain the positive inotropic response to PDE1 inhibition with ITI-214, it was reported that in primary guinea pig myocytes, ITI-214 could enhance myocyte contractility through a PKA-dependent increase in Ca_V_1.2 (L-type calcium channel) conductance without altering SERCA2a or Na^+^/Ca^2+^ exchanger function [5]. Interestingly, PDE1 was found to be localized in close functional proximity of Ca_V_1.2 channels, which are known to reside in caveolin-rich membrane microdomains where NO-GC is also located [5].

ITI-214 is the highly selective inhibitor for PDE1 subtypes; however, it has similar affinities for PDE1A and PDE1C [19]. Therefore, at present, it is not possible to distinguish completely which PDE1 subtype—1A or 1C—plays a role in ITI-214 effects in vivo. In our study, we used the siRNA-mediated silencing of both PDE1 subfamilies to study their relative roles in the regulation of cGMP dynamics in the NO-GC/cGMP microdomain. Despite the fact that PDE1C is the main Ca^2+^/calmodulin-stimulated PDE in myocytes and PDE1A is expressed at very low levels, PDE1A seems to be an important regulator of local cGMP and its effect is amplified by PDE1C. It is possible that both PDE1A and PDE1C, which obviously act in concert to regulate cGMP levels, localize in close proximity to NO-GC, and therefore, the down-regulation of both subfamilies could produce a complete effect. Further studies are needed to study their exact subcellular localization and interaction with macromolecular complexes. Importantly, since PDE1C is a predominant PDE1 subfamily in the human heart and has a higher hydrolytic activity compared to PDE1A hydrolytic activities (about 1/20–1/50) [3], ITI-214 effects on NO/cGMP and CNP/cGMP in myocytes from humans and larger mammals could be mainly due to its inhibition of PDE1C.

However, several studies were performed using PDE1C KO mice showing a connection between PDE1C and the cAMP/PKA signaling pathway but not the cGMP/PKG pathway. Knight et al. [24] showed, in adult mouse cardiomyocytes, that PDE1C deficiency (through *pde1c* gene deletion) or inhibition (with IC86340) attenuated myocyte death and apoptosis, which was linked to cAMP/PKA and PI3K/AKT signaling. Two years later, the same group could identify the specific cAMP signaling pathway modulated by PDE1C and determine the mechanism by which PDE1C is activated. They showed that PDE1C is found in a multiprotein complex together with the transient receptor potential–canonical channel member 3 (TRPC3) and A_2_R. Thereby, PDE1C, activated by TRPC3-derived Ca^2+^, antagonizes A_2_R-cAMP signaling and promotes cardiomyocyte death/apoptosis [25].

Since PDE1C is the main PDE1 isoform expressed in larger mammals, including humans, and the inotropic responses found in the respective studies were PKA-dependent, it was claimed that cAMP is the primary regulated cyclic nucleotide by PDE1. However, the precise localization of PDE1-regulated cAMP microdomains remained uninvestigated, and besides cAMP, PDE1C is known to degrade cGMP with similar affinities. It is possible that PDE1C acts as a potential integrator at the interface of cAMP-, cGMP-, and Ca^2+^-mediated signals in cardiomyocytes [26]. Furthermore, not only PDE1C but also PDE1A, which is >20 times more selective for cGMP, is expressed in larger mammals and humans [4,6]. So far, the regulation of cardiac cGMP by PDE1 has not been directly addressed and the potential role for cGMP remains unanswered.

While in this study, the preinhibition of PDE1 did not affect the amplitude of CNP-induced cGMP-FRET response in resting and paced cardiomyocytes (Figure 2C), it was able to increase the SNP-induced cGMP-FRET response by ~2-fold (Figure 3C). These findings suggest a direct functional role of PDE1 in the local restriction of NO-GC-generated cGMP. However, it needs to be mentioned that, here, we used only the cytosolic FRET sensor red-cGES-DE5 [9] for our FRET measurements. Particulate GCs were localized to the cardiomyocyte membrane. Even though it is known that CNP leads to far-reaching cGMP signals in the cell [27], it would be interesting to investigate the effect of PDE1 inhibition using the cGMP biosensor pmDE5 targeted to the caveolin-rich membrane domain in myocytes [27].

PDE1 is a Ca^2+^/calmodulin-regulated PDE. Interestingly, we saw no significant differences in FRET-response to ITI-214 alone or ITI-214 after CNP-prestimulation between resting and paced myocytes (Figure 2), suggesting the low impact of rapidly changing Ca^2+^ concentrations on pGC-generated cytosolic cGMP. However, investigating the cGMP-pool generated by NO-GC, we saw significantly increased ITI-214 FRET response after SNP prestimulation in paced cells compared to resting cells (Figure 3). This supports our hypothesis that PDE1 plays an important role in the regulation of the NO-GC/cGMP microdomain (Figure 8).

In a study published by our group a few years ago, we investigated the interactions of Ca^2+^ fluctuations during cardiomyocyte contraction with real-time cAMP dynamics in cardiomyocytes expressing the cAMP FRET biosensor Epac1-camps in the cytosol and the subsarcolemmal microdomains [10]. While we could not detect any significant difference between resting and paced cardiomyocytes in the cAMP-FRET response to the PDE1 inhibitor 8-methoxymethyl-3-isobutyl-1-methylxanthine (8-MMX) after isoprenaline treatment, the prestimulation of cells with the direct adenylyl cyclase activator forskolin significantly increased PDE1 contribution to cAMP hydrolysis in paced myocytes compared to resting myocytes. This effect could be mimicked by the preincubation of resting cardiomyocytes with Ca^2+^ elevating agents such as thapsigargin and calcium ionophore, suggesting the calcium-dependent nature of this response [10].

Immunoblot analysis revealed significantly increased PLN phosphorylation at Ser-16 in myocytes treated with a combination of SNP and PDE1 inhibitor but not with SNP alone (Figure 5), which was not PDE3-related (Figure 6). Together with our myocyte contraction data showing that ITI-214 in combination with SNP but not SNP alone has positive lusitropic effects (Figure 4), these results support our FRET data showing that PDE1 is involved in the local restriction NO-GC-generated cGMP.

While in our study, ITI-214 alone was able to significantly increase PLN phosphorylation at Ser-16 (Figure 5), Muller et al. [5] was not able to detect any significant elevation of PLN phosphorylation by ITI-214 alone. There could be several reasons for this. We used mouse cardiomyocytes in our study, whereas Muller et al. used myocytes from guinea pigs, which differ in their PDE1 isoform expression. Furthermore, to probe for PLN phosphorylation, Muller et al. incubated the myocytes for 5 min with drug solutions [5], whereas we chose 10 min as the incubation time in our study.

Further investigations will be needed to show whether the administration of ITI-214 in vivo could produce changes in the cGMP pathway. Importantly, based on our present study, a combination of drugs, such as PDE1C inhibitor and cGMP-elevating drugs, is needed to test therapeutic effects. There is a known danger that the combined application of cGMP-elevating drug and cGMP-PDE inhibitors could produce severe side effects. For example, using sildenafil, targeting PDE5 in blood vessels, together with nitroglycerin (NO/GC/cGMP production also in blood vessels) is not recommended, since these medications together could cause a sharp drop in blood pressure.

However, potentially PDE1C-specific inhibitors could be a good choice for combined application with cGMP-elevating drugs in humans, such as GC activators or stimulators. It has been shown that PDE1C is not expressed in the human aorta [28], and thus, effects of this treatment on blood pressure cannot be expected. On the other hand, PDE1C was shown to be induced in isolated proliferating human smooth muscle cells, which could occur, for example, during the development of atherosclerotic plaque. Therefore, the application of the PDE1C inhibitor under those pathological conditions could also be beneficial. In contrast, PDE1A is expressed ubiquitously in multiple tissues with high level expression in smooth muscle cells. Therefore, specific PDE1A inhibitors cannot be used as cardio-specific drugs [28].

## 5. Conclusions

In conclusion, we have demonstrated a so far unknown role of PDE1 in the regulation of the NO-GC/cGMP microdomain linked to ventricular myocyte contractility. Our data suggest that PDE1 inhibition could be beneficial in combination with cGMP-elevating drugs for the treatment of cardiac diseases.

## Figures and Tables

**Figure 1 cells-12-02759-f001:**
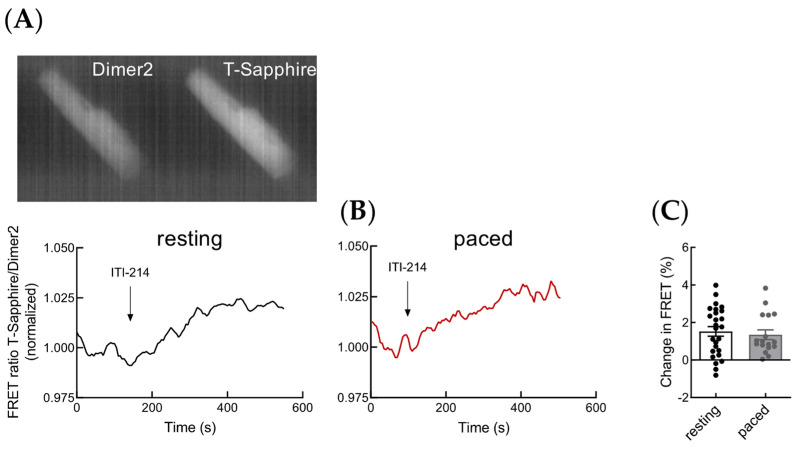
PDE1 inhibition increases cGMP-FRET in murine ventricular myocytes. (**A**,**B**) Representative raw fluorescence cell images in both fluorophore channels of the red-cGES-DE5 sensor and cGMP-FRET responses to PDE1 inhibition with ITI-214 (1 µM) alone. Representative traces showing the effect in resting (**A**) and paced (**B**) myocytes. (**C**) Quantification of cGMP-FRET responses to basal PDE1 inhibition shown in (**A**,**B**). Number of measured cardiomyocytes/mice were as follows: resting, 26/9; paced, 17/7.

**Figure 2 cells-12-02759-f002:**
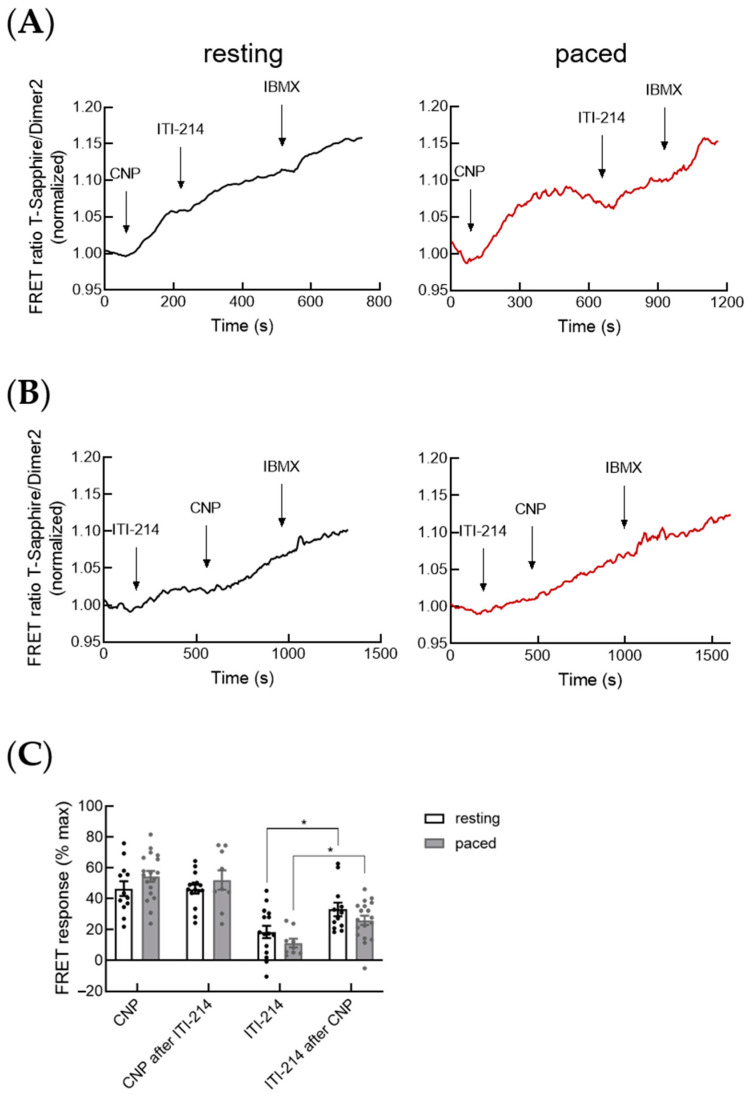
PDE1 is not directly involved in the regulation of the cGMP pool generated by particulate guanylyl cyclase. (**A**) Representative FRET traces for the cGMP-FRET response in resting and paced myocytes to PDE1 inhibition with ITI-214 (1 µM) after stimulation with natriuretic peptide CNP (30 nM) followed by the pan-PDE inhibitor IBMX (100 µM). (**B**) Representative FRET traces for the cGMP-FRET response in resting and paced myocytes treated with ITI-214 (1 µM) followed by CNP (30 nM) and IBMX (100 µM). (**C**) Quantification of the FRET responses shown in (**A**,**B**). Number of measured cardiomyocytes/mice were as follows: CNP resting and ITI-214 after CNP resting = 12/6; CNP paced and ITI-214 after CNP paced = 18/4; ITI-214 resting and CNP after ITI-214 resting = 15/5; ITI-214 paced and CNP after ITI-214 paced = 9/3. Data in (**C**) were analyzed using nested *t*-test, * *p* < 0.05.

**Figure 3 cells-12-02759-f003:**
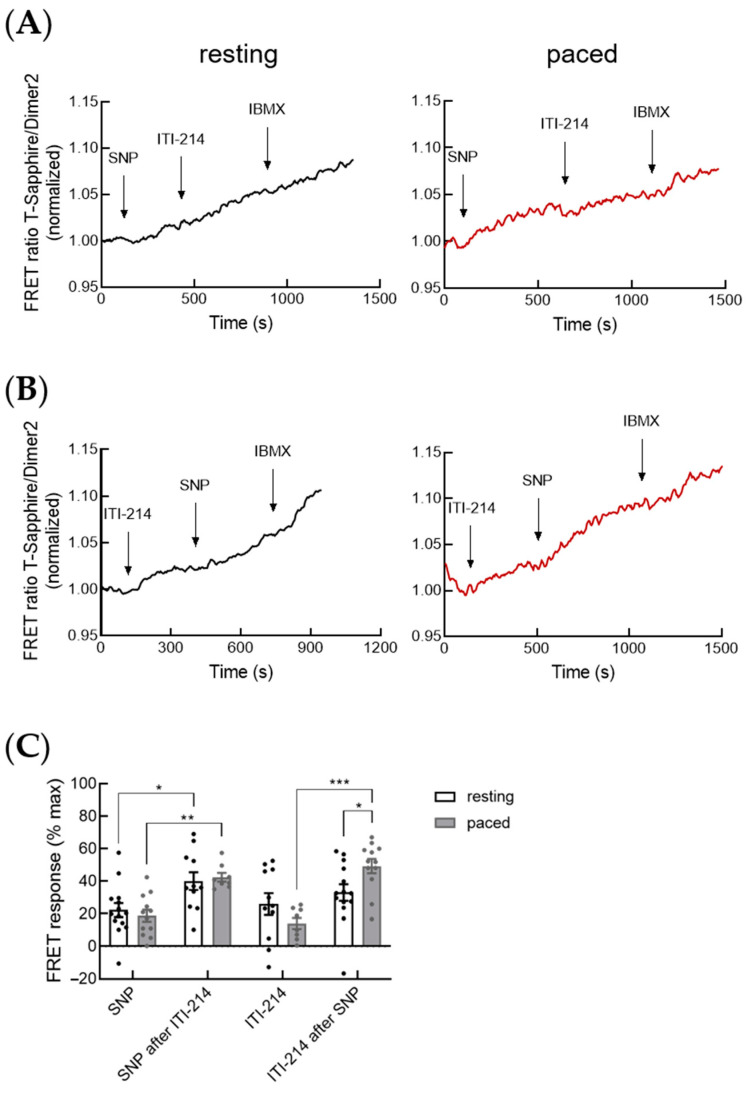
PDE1 is involved in the regulation of the cGMP pool generated by NO-GC. (**A**) Representative FRET traces for the cGMP-FRET response in resting and paced myocytes to PDE1 inhibition with ITI-214 (1 µM) after stimulation with NO-donor sodium nitroprusside (SNP, 50 µM), followed by the pan-PDE inhibitor IBMX (100 µM). (**B**) Representative FRET traces for cGMP-FRET response in resting and paced myocytes treated with ITI-214 (1 µM) followed by SNP (50 µM) and IBMX (100 µM). (**C**) Quantification of FRET responses shown in (**A**,**B**). Number of measured cardiomyocytes/mice were as follows: SNP resting and ITI-214 after SNP resting = 14/4; SNP paced and ITI-214 after SNP paced = 13/5; ITI-214 resting and SNP after ITI-214 resting = 11/4; ITI-214 paced and SNP after ITI-214 paced = 8/4. Data in (**C**) were analyzed via a nested *t*-test, * *p* < 0.05, ** *p* < 0.01, *** *p* < 0.001.

**Figure 4 cells-12-02759-f004:**
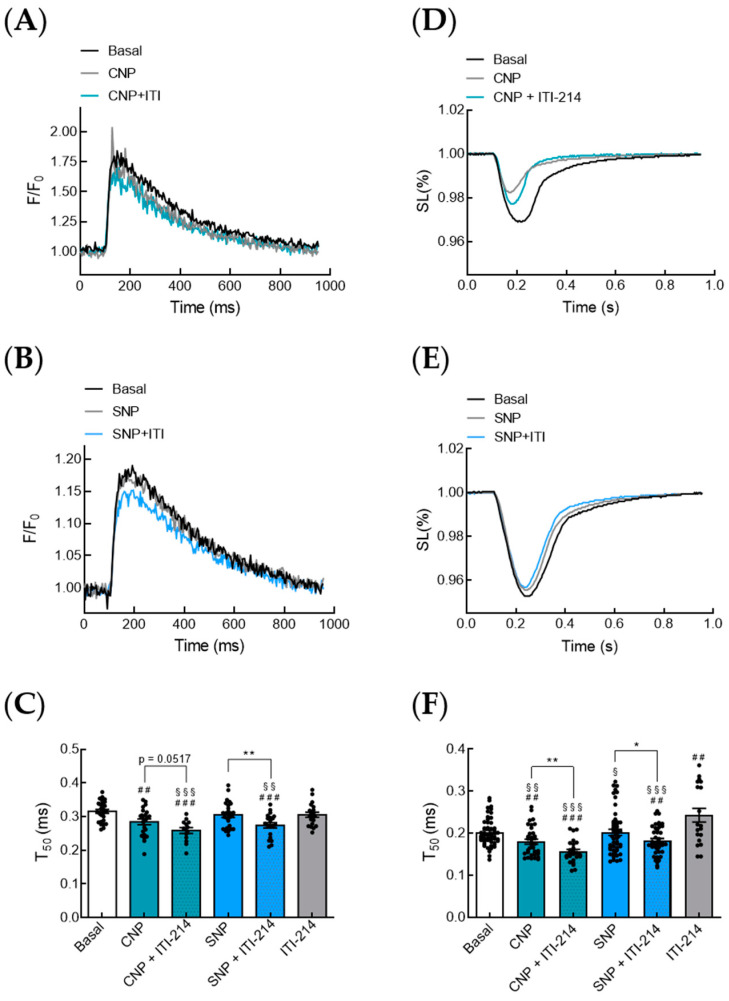
ITI-214 in combination with CNP or SNP shows positive lusitropic effects in isolated myocytes. (**A**) Representative Ca^2+^ transients from cardiomyocytes treated with C-type natriuretic peptide CNP (100 nM) or CNP in combination with PDE1 inhibitor ITI-214 (1 µM). (**B**) Representative Ca^2+^ transients from cardiomyocytes treated with the NO-donor SNP (50 µM) or SNP in combination with the PDE1 inhibitor ITI-214 (1 µM). (**C**) Analysis of Ca^2+^ re-uptake kinetics (T_50_) from experiments shown in (**A**,**B**). (**D**) Representative traces (mean of five cells) of sarcomere shortening from cardiomyocytes treated with natriuretic peptide CNP (100 nM) or CNP in combination with PDE1 inhibitor ITI-214 (1 µM). (**E**) Representative traces (mean of five cells) of sarcomere shortening from cardiomyocytes treated with NO-donor SNP (50 µM) or SNP in combination with PDE1 inhibitor ITI-214 (1 µM). (**F**) Relaxation (T_50_) in isolated myocytes treated with CNP (100 nM) or CNP in combination with ITI-214 (1 µM), SNP (50 µM) or SNP in combination with ITI-214, or ITI-214 (1 µM) alone. Number of measured cardiomyocytes/mice were as follows: basal = 49/5; CNP = 35/5; CNP + ITI-214 = 25/5; SNP = 50/8; SNP + ITI-214 = 41/7; ITI-214 = 19/5. Data in (**C**,**F**) were analyzed using an unpaired *t*-test, * *p* < 0.05, ** *p* < 0,01, ## *p* < 0.01 vs. basal, ### *p* < 0.001 vs. basal, § < 0.05 vs. ITI-214, §§ < 0.01 vs. ITI-214, §§§ < 0.001 vs. ITI-214.

**Figure 5 cells-12-02759-f005:**
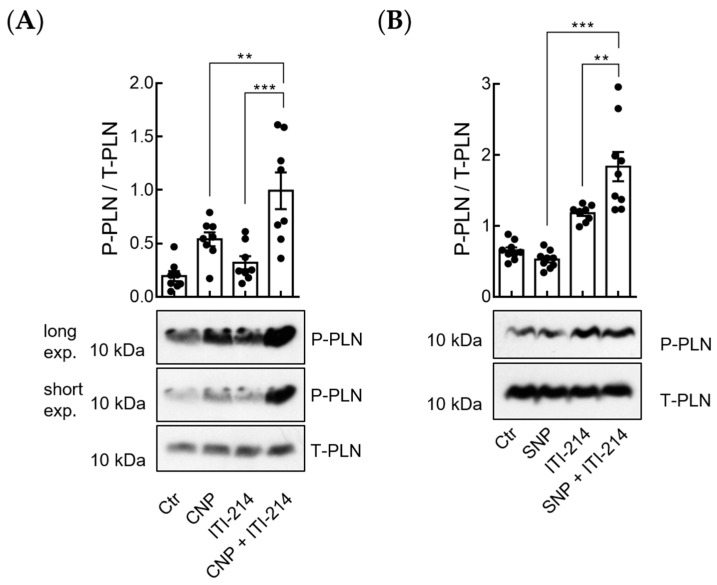
Inhibition of PDE1 amplifies phospholamban (PLN) phosphorylation at Ser-16 upon CNP and SNP stimulation. (**A**) Isolated cardiomyocytes were stimulated with CNP (10 nM), ITI-214 (1 µM), and the combination of CNP (10 nM) + ITI-214 (1 µM). Representative immunoblots and the quantification of phospholamban phosphorylation at Ser-16 (P-PLN). Samples were normalized to total phospholamban (T-PLN). Quantification from *n* = 8 mice is shown. (**B**) Representative immunoblots and quantification of P-PLN in isolated murine myocytes stimulated with SNP (50 µM), ITI-214 (1 µM), and SNP (50 µM) + ITI-214 (1 µM). Samples were normalized to T-PLN. Quantification from *n* = 9 mice is shown. Data in (**A**,**B**) were analyzed using one-way ANOVA followed by Sidak’s multiple comparisons test, ** *p* < 0.01, *** *p* < 0.001.

**Figure 6 cells-12-02759-f006:**
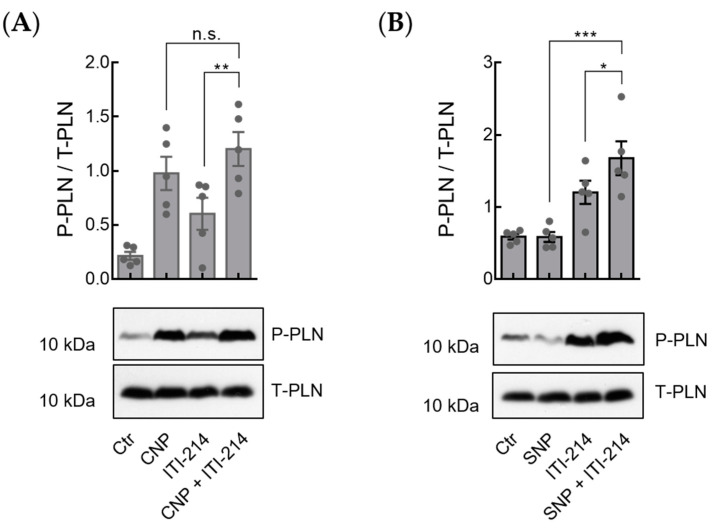
Inhibition of PDE1 amplifies phospholamban phosphorylation at Ser-16 upon CNP and SNP stimulation in PDE3A KO mice. (**A**,**B**) Isolated cardiomyocytes from homozygous PDE3A KO mice (PDE3A+/+/Cre+) were stimulated with CNP (10 nM), ITI-214 (1 µM), and CNP (10 nM) + ITI-214 (1 µM) (A) or SNP (50 µM), ITI-214 (1 µM), and SNP (50 µM) + ITI-214 (1 µM) (B). Representative immunoblots and the quantification of phospholamban phosphorylation at Ser-16 (P-PLN) and total phospholamban (T-PLN). Samples were normalized to T-PLN. Quantification from n = 5 mice for (**A**,**B**) are shown. Data in (**A**,**B**) were analyzed using one-way ANOVA followed by Sidak’s multiple comparisons test, * *p* < 0.05, ** *p* < 0.01, *** *p* < 0.001, n.s.—not significant.

**Figure 7 cells-12-02759-f007:**
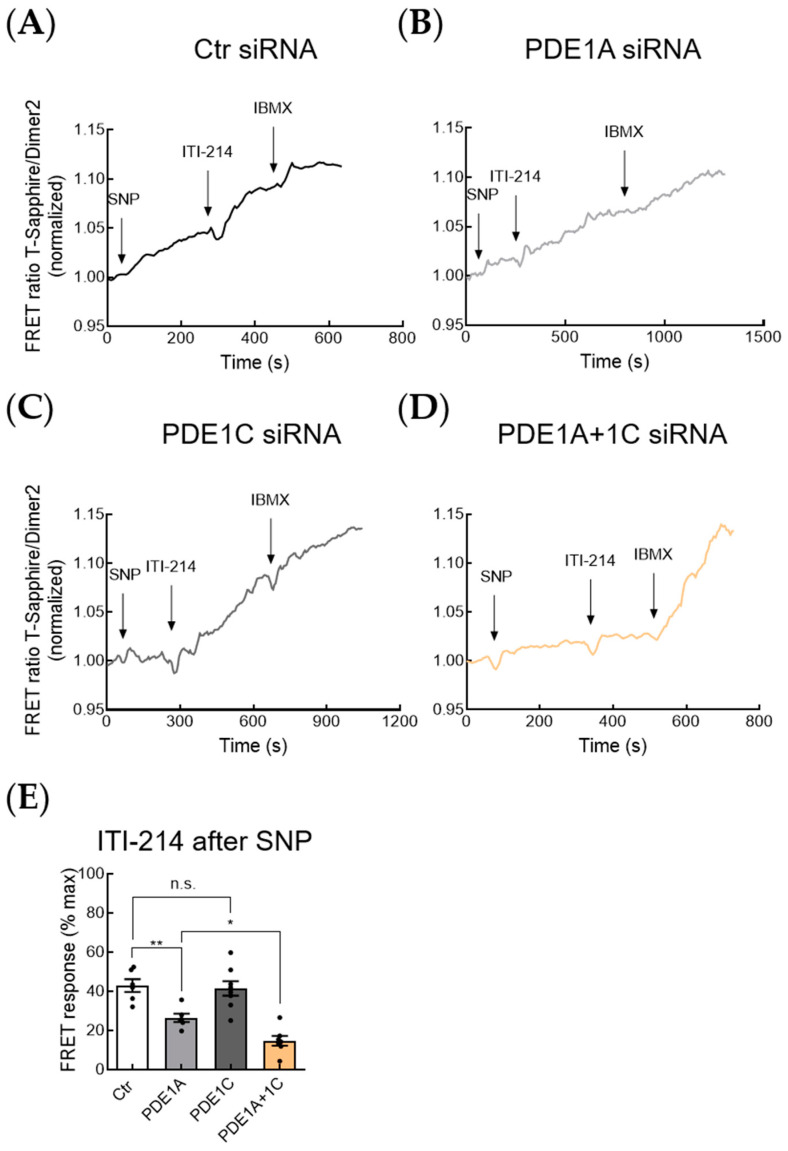
PDE1A and PDE1C in concert regulate the NO-GC/cGMP microdomain. (**A**–**D**) Representative FRET traces for the cGMP-FRET response in resting myocytes after siRNA treatment with Ctr (**A**), PDE1A (**B**), PDE1C (**C**), and PDE1A + 1C (**D**) siRNA to PDE1 inhibition with ITI-214 (1 µM) after stimulation with NO-donor sodium nitroprusside (SNP, 50 µM), followed by the pan-PDE inhibitor IBMX (100 µM). (**E**) Quantification of ITI-214 FRET responses after stimulation with SNP shown in (**A**–**D**). Number of measured cardiomyocytes/mice were as follows: Ctr siRNA = 6/3; PDE1A siRNA = 6/3; PDE1C siRNA = 8/3; PDE1A + 1C siRNA = 7/3. Data in (**E**) were analyzed using one-way ANOVA followed by Sidak’s multiple comparisons test, * *p* < 0.05, ** *p* < 0.01, n.s.—not significant.

**Figure 8 cells-12-02759-f008:**
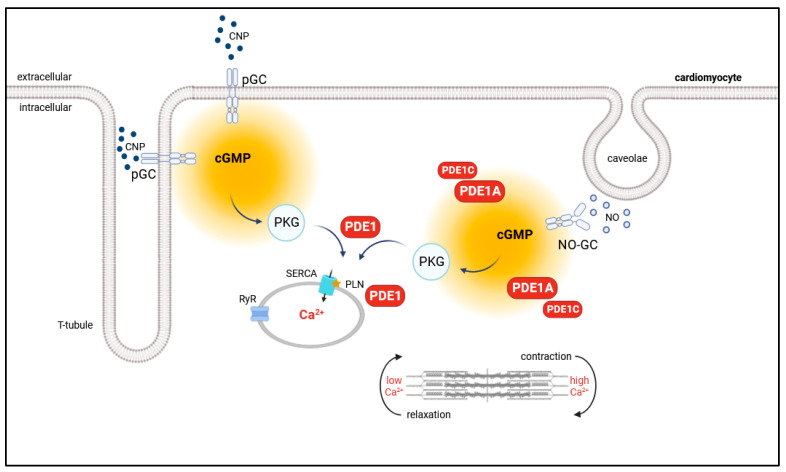
Schematic illustration of the proposed role of PDE1 in the regulation of the NO-GC/cGMP microdomain and mouse ventricular myocyte contractility. Cardiac cGMP is compartmentalized in subcellular microdomains by local pools of PDEs and GC, including NO-GC and pGC. PDE1A in concert with PDE1C compartmentalizes NO-GC-generated cGMP. Additionally, PDE1 controls cGMP/PKG signaling at the sarcoplasmic reticulum, thereby regulating myocyte contractility.

## Data Availability

Data are contained within the article or the Appendix A. Raw data and materials are available from the authors upon reasonable request.

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
