# Peer review of "Role of Phosphodiesterase 1 in the Regulation of Real-Time cGMP Levels and Contractility in Adult Mouse Cardiomyocytes"

_cells, 2023, doi:10.3390/cells12232759_

Round 1
Reviewer 1 Report
Comments and Suggestions for Authors
The manuscript from Bork and colleagues presents findings based on live cell FRET imaging of cardiomyocyte cGMP which infer that different functional compartments of cGMP in response to selective stimulation of sGC particulate GC are regulated by PDE1C.
Results, which are obtained with a rather conventional signalling pharmacology study using specific PDE1 inhibitors, and agonist-effect readouts, are generated by a group with extensive experience on the methods and are technically sound.
I have some concerns, as follows:
- sarcomere length is shown to describe the lusitropic effect of PDE1C inhibition in response to natriuretic peptide of SNP. While mention is made to cytosolic Ca2+ measurement (Fura-2?), these are not adequately represented in the figures.
- The effect on SL is similar in amplitude but - at least in the representative plot - very different in kinetics, with SNP alone shortening the contraction time and SNP+ITI normalising it. How do the authors explain this?
- what is the structural basis of cGMP compartmentation by PDE1C? is intracellular localisation coherent with spatial dynamics of SNP or CNP stimulation, i.e. receptors/sGC/PDE1C?
- Adequate analysis of Ca2+ transient dynamics in response to the cGMP increasing stimuli and PDE inhibitors should be added. This is particularly true when conclusions imply effects on SR Ca2+ reputake
- Figures could be improved:
cells are never shown.
representative Ca2+ traces should be shown (which links to the point stated above, analysis of Ca2+ dynamics should).
a cartoon, explaining the working model, could be added to support purported conclusions, in particular would serve to understand the authors view on cGMP compartmentation.
Author Response
We thank the Reviewer for their positive evaluation of our manuscript and constructive criticism, including important suggestion which were now all addressed during the revision.
Comments and Suggestions for Authors
1. I would suggest to include representative FRET images for Figures 1-3
We thank the Reviewer for this suggestion. However, the traces we are showing (Figure1A, B; Figure 2A, B; Figure 3A, B) are already representative FRET traces from our FRET measurements.
For Figures 5 and 6, it may be more appropriate to do a different statistical analysis [i.e., One-way ANOVA followed by a post-hoc test such as Dunnett’s test]. This would allow you to make direct comparisons among all the groups in comparison to the control.
We fully agree with the Reviewers comment. We have changed the statistics for Figure 5 and 6 and used one-way ANOVA followed by Sidak’s multiple comparisons test (as suggested, also by our GraphPad Prism software) to compare CNP vs CNP + ITI-214 and ITI-214 vs CNP + ITI-214 in Figures 5A and 6A and to compare SNP vs SNP + ITI-214 and ITI-214 vs SNP + ITI-214 in Figures 5B and 6B. Comparison to the control groups was not of our particular interest in these figures, as we wanted to see whether ITI-214 can enforce stimulation with CNP or SNP, so we did not include those comparisons.
- To complement the inhibitor studies (Figure 5), it might be helpful to perform some genetic interference (e.g., siRNA) studies to confirm the role of PDE1 in PLN phosphorylation and that the observed changes are not due to off-target effects.
We fully agree with the Reviewers comment. In order to address this question and also to better understand which PDE1 subfamily – PDE1A or PDE1C (see also question 1 by other Reviewer 2) – is responsible for the regulation of the NO-GC/cGMP microdomain in mouse cardiomyocytes, we selectively silenced PDE1A, PDE1C or both subfamilies together in mouse cardiomyocytes using siRNA. cGMP-FRET measurements showed, that silencing of PDE1A significantly reduced the FRET response to ITI-214 after stimulation with NO-donor sodium nitroprusside (SNP), whereas silencing PDE1C had no significant effect on ITI-214 FRET response alone but significantly increased PDE1A siRNA effect when both subfamilies were knocked down together (new Figure 7). We could perform FRET measurements in single myocytes with PDE1 knockdown after 2 days in culture. However, low viability of mouse myocytes did not allow us to collect sufficient amount of cells for western blot analysis of PLN phosphorylation.
To verify that the siRNAs work for 1A and 1C knockdown we did immunoblot control in transfected adult mouse myocytes which worked for PDE1C (new Supplementary Fig 1A).
Due to low expression and antibody specificity issues for PDE1A (the latter one not detectable with available antibodies in myocytes vs. well detectable in fibroblasts):
we had to use another approach. Instead, we co-trasnfected PDE1A plasmid and siRNAs into HEK293 cells which showed that the used siRNA could reduce PDE1A expression (new Supplementary Fig 1B).
Minor edits:
Line 136: Spelling error should be basal
We thank the Reviewer for indicating this mistake. We have changed this in the text.
Line 141: Should be receptors
We thank the Reviewer for indicating this mistake. We have fixed it in the text.
Line 252: You could say ensure instead of make sure and remove the comma
We thank the Reviewer for indicating this mistake. We have changed this in the text.
Lines 308 and 315: Spelling error, it seems it should be larger
We thank the Reviewer for indicating this mistake. We have changed this in the text.

Reviewer 2 Report
Comments and Suggestions for Authors
1. I would suggest to include representative FRET images for Figures 1-3
2. For Figures 5 and 6, it may be more appropriate to do a different statistical analysis [i.e., One-way ANOVA followed by a post-hoc test such as Dunnett’s test]. This would allow you to make direct comparisons among all the groups in comparison to the control.
3. To complement the inhibitor studies (Figure 5), it might be helpful to perform some genetic interference (e.g., siRNA) studies to confirm the role of PDE1 in PLN phosphorylation and that the observed changes are not due to off-target effects.
Minor edits:
Line 136: Spelling error should be basal
Line 141: Should be receptors
Line 252: You could say ensure instead of make sure and remove the comma
Lines 308 and 315: Spelling error, it seems it should be larger
Comments on the Quality of English LanguageMinor editing suggested. Spelling check recommended.
Author Response
Comments and Suggestions for Authors
In their MS the authors study the role of PDE1 in controlling the cGMP concentration in cardiomyocytes using a selective inhibitor of this PDE. They further interrogate the link of the NOS pathway and show how these 2 pathways converge to control the phosphorylation of PLN and thereby presumably the lusitropic effects of blocking PDE1.
This is a well-carried study with important findings and scientific implications.
There is one issue that is not fully addressed, the isoform that is responsible for the effect. Although the authors discuss why PDE1C is the probable target it is still possible that other forms may participate since the inhibitor is selective to PDE1 but not to distinct forms. Can the author try to selectively silence PDE1C? at least in cardiac cell lines strengthening by any other experimental strategy the notion that it is PDE1C?
We fully agree with the Reviewers comment. A similar suggestion was made by Reviewer 1 (please see above). In order to address the question which PDE1 subfamily – PDE1A or PDE1C – is the responsible for the regulation of the NO-GC/cGMP microdomain in mouse cardiomyocytes, we used an siRNA mediated approach to selectively silence PDE1A, PDE1C or both subfamilies together in mouse cardiomyocytes. cGMP-FRET measurements showed, that silencing PDE1A significantly reduced the FRET response to ITI-214 after stimulation with NO-donor sodium nitroprusside (SNP), whereas silencing PDE1C had no significant effect on ITI-214 FRET response (new Figure 7, see previous page).

Reviewer 3 Report
Comments and Suggestions for Authors
In their MS the authors study the role of PDE1 in controlling the cGMP concentration in cardiomyocytes using a selective inhibitor of this PDE. They further interrogate the link of the NOS pathway and show how these 2 pathways converge to control the phosphorylation of PLN and thereby presumably the lusitropic effects of blocking PDE1.
This is a well-carried study with important findings and scientific implications.
There is one issue that is not fully addressed, the isoform that is responsible for the effect. Although the authors discuss why PDE1C is the probable target it is still possible that other forms may participate since the inhibitor is selective to PDE1 but not to distinct forms. Can the author try to selectively silence PDE1C? at least in cardiac cell lines strengthening by any other experimental strategy the notion that it is PDE1C?
Author Response
The manuscript from Bork and colleagues presents findings based on live cell FRET imaging of cardiomyocyte cGMP which infer that different functional compartments of cGMP in response to selective stimulation of sGC particulate GC are regulated by PDE1C.
Results, which are obtained with a rather conventional signalling pharmacology study using specific PDE1 inhibitors, and agonist-effect readouts, are generated by a group with extensive experience on the methods and are technically sound.
I have some concerns, as follows:
- sarcomere length is shown to describe the lusitropic effect of PDE1C inhibition in response to natriuretic peptide of SNP. While mention is made to cytosolic Ca2+ measurement (Fura-2?), these are not adequately represented in the figures.
We fully agree with the Reviewers comment and apologies for this major omission. We have now included some representative Ca2+ transients in Figure 4 (A, B) and data analysis on Figure 4 ©.
- The effect on SL is similar in amplitude but - at least in the representative plot - very different in kinetics, with SNP alone shortening the contraction time and SNP+ITI normalising it. How do the authors explain this?
We thank the Reviewer for this comment. We have changed the representative traces of sarcomere shortening for SNP to more appropriate ones (Figure 7 D).
- what is the structural basis of cGMP compartmentation by PDE1C? is intracellular localisation coherent with spatial dynamics of SNP or CNP stimulation, i.e. receptors/sGC/PDE1C?
We thank the Reviewer for this questions. Unfortunately, the specificity of the available antibodies was not good enough to allow for reliable PDE1 staining in our cell system. It is expected that the signalling compontens such as sGC and PDE1C are assembled in signalosomes in close proximity. sGC is known to be localized close to caveolin rich plasma membrane of myocytes (as shown by us previously PMID ). Ref 5 (Muller et al) also shows that PDE1 regulated L-type calcium current (this channel is known to reside in caveolin rich plasma membrane microdomains) suggesting their close proximity in addition to the data published by Liu and Marx (PMID: 31969708) using proximity proteomics. We have included some of these consideration in our discussion and added also a summary slide in Figure 8 depicting this potential scenario.
- Adequate analysis of Ca2+ transient dynamics in response to the cGMP increasing stimuli and PDE inhibitors should be added. This is particularly true when conclusions imply effects on SR Ca2+ reuptake
Done, as described above in question 1.
- Figures could be improved:
Cells are never shown
We thank the Reviewer for this suggestion and now show some representative cell images in Figure 1A.
Representative Ca2+ traces should be shown (which links to the point stated above, analysis of Ca2+ dynamics should).
Done, as described above in question 1.
A cartoon, explaining the working model, could be added to support purported conclusions, in particular would serve to understand the authors view on cGMP compartmentation.
We thank the Reviewer for this suggestion. We have made a cartoon explaining our working model, please see New Figure 8.
